# Allogeneic TCRαβ deficient CAR T-cells targeting CD123 in acute myeloid leukemia

Mayumi Sugita [1], Roman Galetto[2], Hongliang Zong[1], Nathan Ewing-Crystal[1], Vicenta Trujillo-Alonso [1], Nuria Mencia-Trinchant[1], Winnie Yip[1], Stephanie Filipe[2], Celine Lebuhotel[2], Agnès Gouble[2], Duane C. Hassane[1], Julianne Smith[2], Gail J. Roboz[1✉] & Monica L. Guzman [1✉]

Acute myeloid leukemia (AML) is a disease with high incidence of relapse that is originated and maintained from leukemia stem cells (LSCs). Hematopoietic stem cells can be distinguished from LSCs by an array of cell surface antigens such as CD123, thus a candidate to eliminate LSCs using a variety of approaches, including CAR T cells. Here, we evaluate the potential of allogeneic gene-edited CAR T cells targeting CD123 to eliminate LSCs (UCART123). UCART123 cells are TCRαβneg T cells generated from healthy donors using TALEN® gene-editing technology, decreasing the likelihood of graft vs host disease. As safety feature, cells express RQR8 to allow elimination with Rituximab. UCART123 effectively eliminates AML cells in vitro and in vivo with significant benefits in overall survival of AML-patient derived xenograft mice. Furthermore, UCART123 preferentially target AML over normal cells with modest toxicity to normal hematopoietic stem/progenitor cells. Together these results suggest that UCART123 represents an off-the shelf therapeutic approach for AML.

[1] Division of Hematology and Oncology, Department of Medicine. Weill Cornell Medical College, New York, NY, USA. [2] Cellectis SA, Paris, France. ✉email: gar2001@med.cornell.edu; mlg2007@med.cornell.edu

Acute myeloid leukemia (AML) is the most common type of acute leukemia in adults and accounts for the largest number of annual deaths from leukemia in the United States[1]. The vast majority of patients relapse despite achieving complete remission with intensive consolidation chemotherapy regimens or stem cell transplantation[1]. Relapse can be attributed to a relatively rare, chemotherapy-resistant subpopulation of cells known as leukemia-initiating cells, or leukemia stem cells (LSCs), that can self-renew, proliferate, and differentiate into leukemic blasts. Patients who present with a higher proportion of phenotypically defined LSCs (CD34+ CD38−) demonstrate significantly poorer relapse-free survival than patients with low proportions of LSCs[2]. Also, a higher proportion of LSCs at diagnosis is highly predictive of minimal residual disease (MRD), suggesting that LSCs are significant contributors to MRD[3]. Importantly, it has been shown that while many chemotherapeutic agents can kill leukemia blasts, no chemotherapeutics routinely used in AML therapy can eliminate LSCs[4]. Together, these data suggest that LSCs play a critical role in disease relapse, cannot be ablated by current therapies, and are a highly relevant therapeutic target in AML.

Chimeric Antigen Receptor (CAR) T-cell therapy represents a major advance in the treatment of hematologic malignancies, as demonstrated by the high response rates and improved survival in diseases such as acute lymphoblastic leukemia (ALL), non-Hodgkin lymphoma and chronic lymphocytic leukemia[5]. CARs are artificial T-cell receptors composed of a monoclonal-antibody (mAb) derived, extracellular-antigen-binding domain fused to an intracellular-TCR signal transduction region. CAR-engineered T-cells can specifically target cancer cell surface antigens by exploiting the antigen-binding properties of monoclonal antibodies to eliminate malignant cells through activation of T-cell-mediated effector functions. CAR-engineered T cells for the treatment of AML are being developed targeting antigens such as CD33, CLL-1, folate receptor β (FRβ), and CD123[6–9].

CD123, the interleukin-3 receptor alpha chain (IL-3Rα), is aberrantly expressed in AML and other hematologic malignancies, such as blastic plasmacytoid dendritic cell neoplasm (BPDCN) and ALL[10,11]. Elevated CD123 expression has been described on normal plasmacytoid dendritic cells and basophils; at lower levels on monocytes, eosinophils and myeloid dendritic cells; and on subsets of hematopoietic progenitor cells and endothelial cells[11,12]. Importantly, CD123 is highly expressed on AML blasts and LSCs, with minimal expression on normal hematopoietic stem cells (HSCs)[13]. Importantly, CD123 has been reported in CD34+ CD38-Lin- populations from healthy bone marrow (BM) and cord blood (CB) donors, with intensities of expression significantly lower than their AML counterparts (such differences were not observed for other myeloid markers such as CD13 and CD33)[14]. Additional studies have also reported that the expression of CD123 in leukemic blasts is significantly higher than in normal myeloid progenitors[15]. Therefore, targeting CD123 offers the potential to selectively eliminate LSCs and represents an attractive therapeutic strategy in AML.

To date, the generation of CAR-T cells has been mainly dependent on autologous cells[16], but this process is limited by significant clinical and logistical barriers[5,17]. Thus, an "off the shelf", allogeneic approach to CAR-T therapy is an attractive solution to provide immediate and guaranteed availability of the cell therapy product for all patients, possibly with greater cost-effectiveness. However, a potential limitation of allogeneic adoptive immunotherapy approaches is that recognition of MHC disparities between donor and recipient through the donor cell's T-cell receptor (TCR) αβ complex may lead to donor T-cell activation/proliferation and the development of graft-versus-host disease (GvHD). Also, the optimal methods and level of lymphodepletion to facilitate allogeneic CAR-T expansion and persistence are unknown[18].

In the present work, we generate allogeneic chimeric antigen receptor (CAR) T cells using T-cells from third-party healthy donors, to produce "universal" CAR T-cells targeting CD123 (UCART123). UCART123 does not express a TCRαβ, having undergone a disruption of the TCRα constant gene (TRAC) and elimination of remaining non-edited TCRαβ-positive cells thus, minimizing the possibility for GvHD[19]. In addition, the CAR construct is co-expressed with RQR8 as a safety feature, to allow elimination with Rituximab. We test the activity of UCART123 in vitro and in vivo using primary AML samples, normal bone marrow (BM) and cord blood (CB) cells, patient-derived xenografts (PDX), normal BM (nBM) humanized xenograft (Hu-X) models, and a competitive nBM/AML xenograft model. Our data show that UCART123 is able to eliminate LSCs and preferentially eliminate AML versus normal cells in vitro and in vivo. UCART123 is a ready to use, off-the-shelf, allogeneic, engineered CAR T-cell product that could offer logistical and efficacy advantages over currently available autologous approaches.

## Results

**Generation of universal CAR T cells targeting CD123**. To specifically target CD123 on AML, we utilized a CAR construct that contains a scFv derived from a murine hybridoma, a CD8 hinge and transmembrane domain, and a cytoplasmic tail composed of 4-1BB co-stimulatory domain and CD3zeta signaling domain. In addition, the lentiviral vector cassette has been designed to co-express RQR8, as safety switch, (through a 2A peptide linker) under the control of an EF1α promoter (detailed in Fig. 1a). RQR8 is an artificial cell surface protein containing two CD20 epitopes that allows depletion of RQR8-expressing cells through complement-mediated cell killing (CDC) and antibody-dependent cell-mediated cytotoxicity (ADCC) in the presence of rituximab (Fig. 1a)[20]. Furthermore, to generate "off the shelf" universal CAR T-cells or UCARTs, TALEN® gene editing technology was used to inactivate the TRAC gene. Inactivation of the TRAC gene prevents cell surface expression of the T-cell receptor (TCR) αβ complex, eliminating TCR-mediated recognition of histocompatibility antigens that can lead to GvHD[21].

We first tested anti-leukemia activity of UCART123 against leukemia cell lines (Fig. 1b and Supplementary Fig. 1a) and found specific cytotoxicity against a CD123-positive AML cell lines (MOLM13, THP1, KG-1, Kasumi-6, and MV4;11) but not against a CD123-negative cell line (Jurkat). Furthermore, robust antigen-specific cytokine production was observed when incubated with MOLM13 cells, but not when incubated with the negative control Jurkat cells (Fig. 1c). The UCART123 activity against leukemia cells was comparable to CART123 cells. Supplementary Table 2, shows the specific characteristics of the UCART123 cells used for the study. It is important to note the high efficiency of the inactivation of TRAC ( < 1% TCR αβ+; Supplementary Fig. 1b) and the high enrichment of CAR + cells (>90%). These features allowed for an efficient in vivo anti-leukemic activity was also observed for UCART123 in NSG mice engrafted with luciferase-expressing MOLM13 cells and treated with UCART123 (Fig. 1d–f). These observations are consistent with the reports for UCART19[22]. In addition, we evaluated the ability of rituximab to eliminate UCART123 (since the lentiviral construct encodes also the RQR8 gene). We found that treatment with 10 mg/kg of rituximab starting 7 days after animals received UCART123 cells resulted in disease progression and decrease in detectable CAR T cells, in contrast with the UCART123 mice that only received vehicle control and remained disease free (Fig. 1h, g

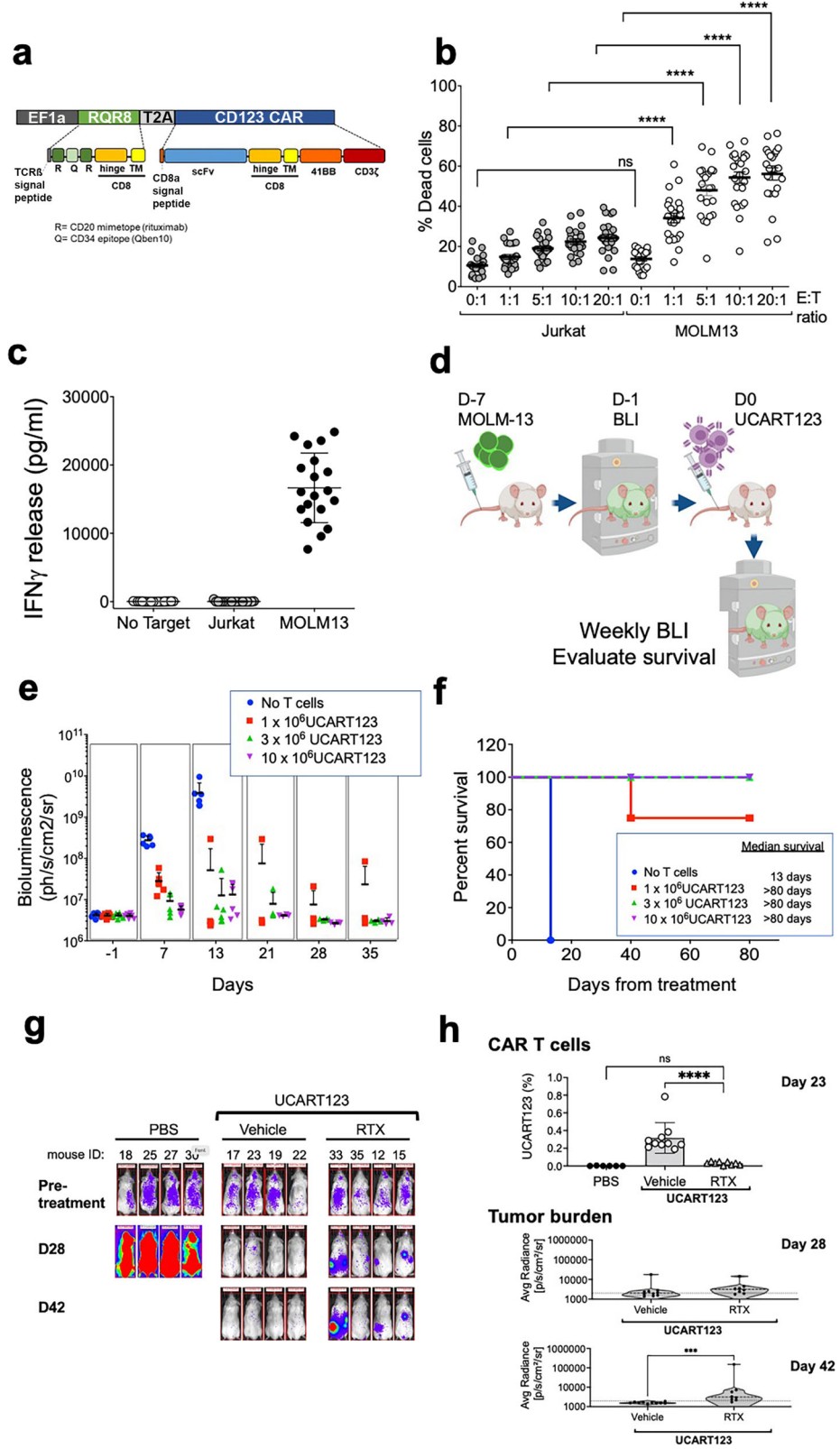

and Supplementary Fig. 1c, d). To confirm that UCART123 were effectively eliminated by rituximab, we re-challenged the animals with leukemia cells (MOLM13). In this study, mice treated with rituximab had significantly lower overall survival and more rapid disease progression compared with UCART123-pretreated mice that received vehicle and no Rituximab (Supplementary Fig. 1e, f), demonstrating that no residual UCART123 were present after

rituximab treatment. Importantly, UCART123 treatment did not induce GvHD (Supplementary Fig. 1g, h); we found that animals treated with UCART123 (99.8% TCRαβ-) did not show body weight loss unlike animals that received the unmodified T cells (NTD) from the same donor (8.9% TCRαβ-). All mice injected with TCRαβ-positive unmodified T-cells showed treatment-related changes consistent with GvHD (weight loss and

**Fig. 1 Evaluation of cytotoxicity by UCART123 against MOLM13 (AML cell line) expressing CD123 in vitro and in vivo. a** CAR construct containing a scFv derived from an anti-CD123 murine antibody linked to a CD8-derived hinge/transmembrane domain, as well as 4-1BB and CD3zeta signaling domains. The CAR is co-expressed with RQR8, an artificial cell surface protein containing two CD20 epitopes recognized by Rituximab. **b** CD123-negative cell line, Jurkat, and CD123-positive AML cell line, MOLM13, were co-cultured with UCART123 with indicated various E:T ratios for 24 h. UCART123 induced significant cell death in MOLM13 at all E:T ratios, but not in Jurkat cells. Each symbol represents independent experiments ($n = 23$) and bar represents the average with the SD. ****$p < 0.0001$ **c** IFN-γ levels in supernatants were measured after 24 h when UCART123 was co-cultured with MOLM13 and Jurkat cells. Each symbol represents independent experiments ($n = 19$) and bar represents the average with the SD. ****$p < 0.0001$. **d** NSG mice were injected with MOLM13 cells expressing luciferase (MOLM13) and were treated with UCART123 at different cell doses on day 7 post injection of MOLM13. Leukemia burden was monitored weekly with bioluminescence imaging (BLI) and survival was evaluated. Diagram created with biorender.com **e** Weekly tracking of leukemia burden by BLI. UCART123-treated mice efficiently eliminated leukemia (except one mouse in the cohort treated with the lowest dose of UCART123), $n = 6$. **f** UCART123-treated mice achieved significant longer survival compared to control ($p = 0.0027$ for each of the UCART123-treated cohorts compared to control, log-rank test). Each symbol represents one mouse. **g** MOLM13 engrafted mice were treated with PBS, UCART123, or UCART123 followed by rituximab (RTX). Representative BLI images at pre-treatment, day 28 and day 42 measured are shown for each group. **h** top, Frequencies of UCART123 (%) in mononuclear cells in peripheral blood on day 23 post UCART123 treatment were measured with flow cytometry. Each symbol represents one mouse and bar represents the average with the SD. ****$p < 0.0001$, one-way ANOVA. bottom, Average radiance measured with BLI on day 28 and day 42 are shown, Each symbol represents a mouse in the cohort (UCART123/vehicle $n = 10$; UCART123/RTX $n = 8$) and the bar represents the average with the SD. ***$p < 0.001$, Mann–Whitney test. Source data are provided as a Source Data file.

histopathological changes). These results offer confidence that UCART123 is unlikely to induce a severe GvHD when administered to patients. In addition, UCART19, a product with similar features (same TALEN-induced TRAC knock-out) has not shown severe GVHD in clinical studies[23].

**UCART123 exhibits cytotoxic activity against primary AML samples with minimum effect against normal hematopoietic progenitor cells.** To further characterize the anti-leukemia activity of UCART123, we assessed the cytotoxic effect of UCART123 against primary AML samples using multi-parameter flow cytometry at 24 h after co-cultures were established at different effector to target ratios (E:T). Importantly, we found over 70% cell death in leukemia cells that were co-cultured for 24 h with UCART123, at E:T ratios as low as 0.5:1 (Fig. 2a). In contrast, control TCRαβ KO T-cells (non-transduced, CAR-negative, TALEN® treated T-cells) induced significantly less cell death when co-cultured with AML cells. Next, we evaluated the ability of UCART123 to produce cytokines when exposed to primary leukemia samples. The levels of IFN-γ and other cytokines (interleukins 2, 4, 5, 6, 9, 10, 13, 17A,17F, 21, 22, and TNF-α) were evaluated in supernatants obtained from 1:1 (E:T) ratio co-cultures after 24 h. Although the cytokine induction profile was variable across AML samples, the highest changes observed were an increased production (average over 20x) of IFN-γ, IL-2, IL-5, IL-13, and TNF-α. No changes were observed in IL-6, IL-17A and IL-17F when comparing UCART123 to TCRαβ KO T-cells (Fig. 2b).

The cytotoxicity of UCART123 against normal hematopoietic cells was evaluated using cord blood (CB) samples from healthy donors. Cytotoxicity in liquid cultures evaluated at 24 h was, on average, less in CB samples ($n = 6$) than in AML samples: 3.9%, 8.5%, and 9.2% for 5:1, 1:1, and 0.5:1 E:T ratios respectively (Fig. 2c). In addition, to determine selectivity of the UCART123 towards AML cells, we evaluated UCART123 cell activity against a co-culture of CB (stained with CFSE; $n = 3$) and a primary AML sample (AML37 stained with cell trace violet; CTV) and evaluated cytotoxicity at 24 h. We found significantly less cell death in CB cells, compared to AML cells (Fig. 2d), and the levels of cell death observed in CB cells were similar to those seen in Fig. 2c when UCART123 cells were co-cultured just with CB cells albeit with more sample-to-sample variability. To determine the effect of UCART123 and non-transduced TCRαβ KO T-cells in normal and leukemia stem/progenitor populations, we performed colony-forming assays in primary AML samples and normal CB cells. Cells were plated 4 h after setting up the co-cultures. We

found a significant decrease in colony-forming units (CFU) in primary leukemia samples after treatment with UCART123, suggesting effective elimination of leukemic progenitor cells at an E:T ratio of 1:1 (Fig. 2e). As expected, a reduction in myeloid colony formation was observed when treating CB samples with UCART123, but only at the highest E:T ratio (5:1) (mean 44%; $N = 10$) (Fig. 2f and see Supplementary Fig. 2a for TCRαβ KO T-cells). However, less toxicity to myeloid progenitors was observed at lower ratios (1:1 and 0.5:1), suggesting that normal hematopoietic progenitors can be spared at lower E:T ratios at which leukemic progenitors can still be efficiently eliminated. Furthermore, we performed secondary replating for CB cells at 1:1 E:T ratios and found no significant change in CFU ($N = 3$; Supplementary Fig. 2b). Taken together, these results show a potent activity of UCART123 against primary AML progenitor cells, and a significantly reduced toxicity of UCART123 against myeloid progenitors in a dose-dependent manner.

**UCART123 targets AML cells in vivo and results in improved overall survival.** To evaluate the in vivo anti-leukemia activity of UCART123, patient-derived xenografts (PDX) from primary AML samples were established. TCRαβ KO cells were used as controls, and different doses of UCART123 were tested. First, we tested higher UCART123 doses as performed in cell line xenografts, as well as other reported studies[24–26]. Specifically, 3 and 10 million cells were evaluated in these initial experiments and animals were killed 3 weeks after the administration of UCART123 to assess leukemic burden and CAR T-cell persistence in the bone marrow (BM) (Supplementary Fig. 3). Strikingly, we found that treatment with 3 and 10 million UCART123 resulted in complete elimination of AML cells within 3 weeks (Supplementary Fig. 3a, b). At this timepoint, all human cells detected in the animals were UCART123 cells (Supplementary Fig. 3c). Since cell numbers were not sufficient to perform secondary transplant assays to assess leukemia-initiating capacity, we proceeded next to evaluate the impact of UCART123 in overall survival (OS) and incidence of relapse.

Since the in vitro cytotoxicity data suggested potent activity at low E:T ratios (0.5:1) and the initial PDX data showed very potent in vivo anti-leukemia activity, we proceeded to evaluate the impact of UCART123 in OS and relapse in PDX-AML mice models at lower doses if UCART123 cells. The new experimental cohorts were established for AML37 and AML2 (Fig. 3a). UCART123 doses of 1 million (PDX-AML2) and 2.5 million (PDX-AML2 and PDX-AML37) were tested and compared with 2.5 million TCRαβ KO or 60 mg/kg Ara-C. The treatment of the

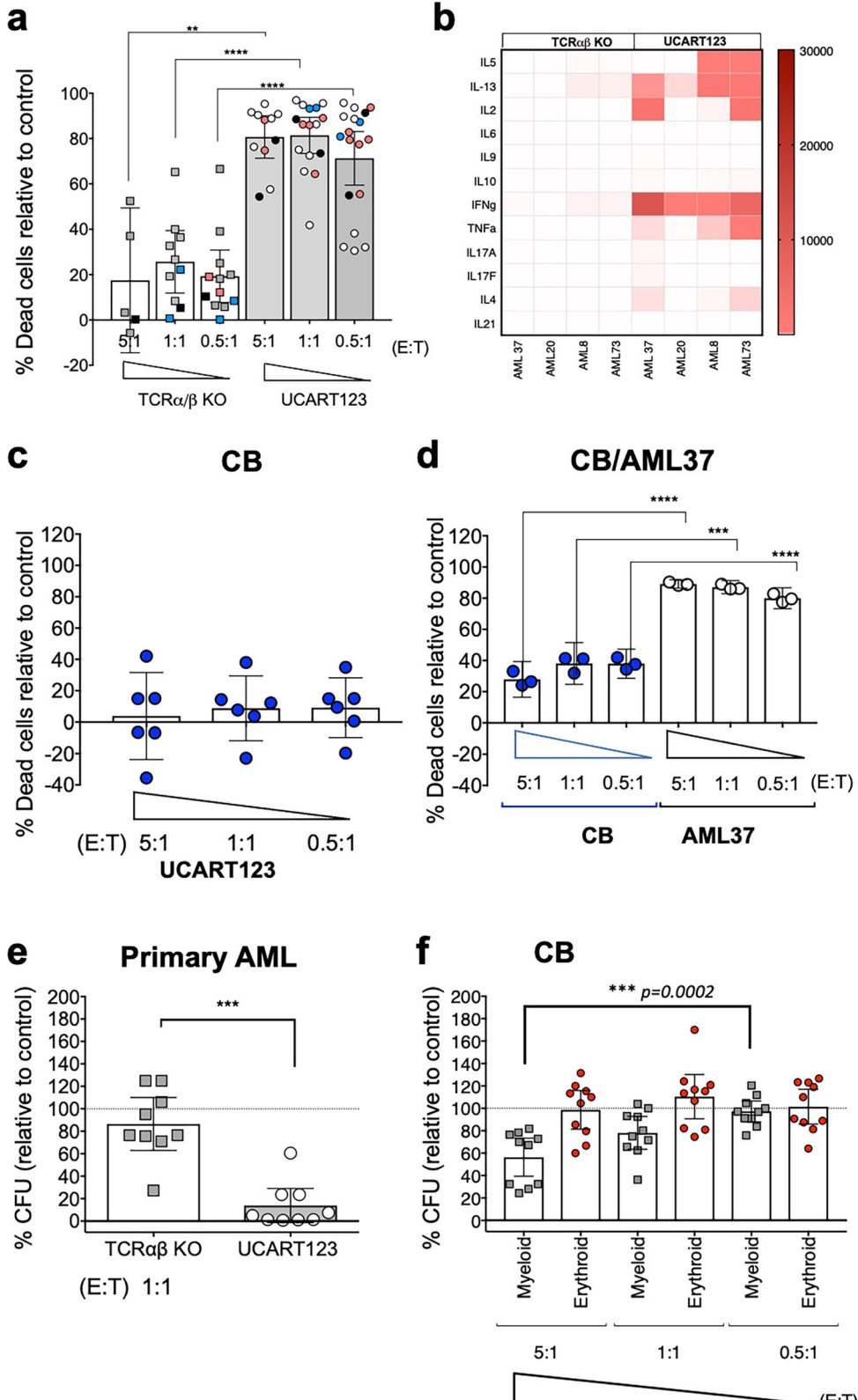

PDXs was started once leukemia was established in the mice and mice were then randomized to the different treatment arms (see Fig. 3a). We found that for both PDX-AML2 and PDX-AML37, treatment with any dose of UCART123 significantly improved OS when compared to saline, TCRαβ KO, or Ara-C treatment controls (Fig. 3b, c). For the PDX-AML37 experiment (Fig. 3c), two animals from each cohort were killed at day 170; of note, the

PBS and Ara-C mice appeared sick at this time point. We obtained peripheral blood (PB) and also harvested BM and spleen. This analysis confirmed that, as observed in PB, UCART123-treated mice did not have leukemia cells in the BM or spleen (Supplementary Fig. 3e). In addition, PDX-AML37 also showed that UCART123-treated mice had about 3 percent of CD123CAR + T-cells in the BM. The rest of the animals were

**Fig. 2 UCART123 inhibited CD123 + primary AML remarkably and selectively with minimum toxicity to normal hematopoietic cells in vitro.** Primary AML cells and cord blood (CB) cells were cultured alone, or co-cultured with TCRαβ KO T cells or UCART123 cells at different effector to target ratios (E:T) in vitro. Cell viabilities were assessed by flow cytometry at all E:T ratios and cytokines released in supernatants were measured after 24 h co-culture at 1:1 E:T ratio. **a** Percent of dead cells relative to untreated control at 24 h after co-culture with TCRαβ KO (white columns) or UCART123 (gray columns) under indicated E:T ratios are shown here. Each symbol represents the mean of replicates of a primary AML sample, $n = 5$ (5:1 ratio), $n = 10$ (1:1 ratio), $n = 11$ (0.5:1 ratio). Samples tested independently more than one time are represented in colors, AML37(red), AML17(blue), and AML95(black). Bar represents the average with the SD. ** $p = 0.0013$, **** $p < 0.0001$ two-tailed t-test. **b** Cytokine levels in supernatants after 24 h co-culture at all E:T ratios were measured. The average concentrations (pg/mL) of each cytokine are presented in the heatmap, ranging 0–30,000. **c** Percent of dead cells of cord blood (CB, $n = 6$) cells relative to untreated control at 24 h after co-culture with UCART123 under indicated E:T ratios. Each symbol indicates one well of replicates. Bar presents the mean with the SD. **d** Percent of dead cells of CB or primary AML at 24 h when mixed cells of CB ($n = 3$) and a single primary AML (AML37) were co-cultured with UCART123 at 5:1, 1:1, and 0.5:1 E:T ratios to assess selectivity of UCART123. UCART123 induced significant less cell death in CB cells compared to cell death in AML37 at all E:T ratios. Each symbol indicates the mean of one CB or AML in replicates. *** $p = 0.0001$, **** $p < 0.0001$. **e** Percent colony-forming units (CFU) relative to control in colony-forming assays of primary AML ($n = 9$) cells plated after 4 h co-culturing with TCRαβ KO or UCART123 at 1:1 E:T ratio. Each symbol represents one well in replicates. Bar represents the mean with the SD. *** $p = 0.0002$ paired two-tailed t-test f, Percent CFU relative to control in colony-forming assays of CB ($n = 10$) cells plated after 4 h co-culturing with UCART123 or TCRαβ KO (see Supplementary fig. 2a) at 5:1, 1:1, and 0.5:1 E:T ratios. Myeloid (gray) and erythroid (red) colonies were evaluated separately. Each symbol represents one well in replicates and bar represents the mean with the SD. **** $p < 0.0001$, *** $p < 0.001$, ** $p < 0.01$, one-way ANOVA. Source data are provided as a Source Data file.

kept alive to determine OS. The experiment was terminated at day 204 from leukemia engraftment (161 from T-cell injection). Analysis of the killed mice showed that UCART123-treated mice were all disease free. In the PDX-AML2 experiment (Fig. 3b), UCART123-treated mice also demonstrated significant improvement in overall survival for both doses tested (1 and 2.5 million). At day 221 from engraftment, all animals were killed and the BM was harvested. All mice from the 2.5 million cohort remained disease free, with only one animal having detectable UCART123 by flow cytometry (Supplementary Fig. 3f). We noted that some, but not all, of animals treated with doses of 1 million were relapsing, suggesting that it was likely that UCART123 were not persisting in all mice. Since, flow cytometry does not have the sensitivity to identify and quantitate the persistence of UCART123 and/or residual disease, we proceeded to develop a more sensitive assay.

Because AML2 carries a mutation in NPM1, we developed a multiplex droplet digital PCR (ddPCR) assay to facilitate monitoring of measurable residual disease (MRD) in PDX-AML2. The ddPCR assay shows approximately 5x higher sensitivity than flow cytometry, and can simultaneously detect NPM1 mutant transcripts, UCART123 transcripts, and house-keeping controls for both human and mouse cells (ABL1 transcripts)[27,28]. The ddPCR assay revealed that the relapsing mice had detectable, and overtime increasing, residual disease with lower levels of CART123 transcripts (Fig. 3d, top panels). In contrast, animals with persisting, or overtime increasing, levels of CART123 transcripts showed a marked decrease of NPM1 transcripts and remained without overt disease relapse (Fig. 3d, bottom panels). All relapsing animals presented CD123 + disease, suggesting that the reason of relapse was likely due to lack of persistence of UCART123. This was confirmed when the relapsing mice were able to respond to a second infusion of UCART123 cells. Observations were reproduced in a second experiment with PDX-AML2 (1 and 5 million treatments). Animals from the UCART123 5 million cohort were re-challenged with AML2 cells at day 180 from UCART123 treatment and most animals remained disease free, demonstrating that UCART123 was still active (Supplementary Fig. 3e).

In summary, we found that treatment with UCART123 in all PDX-AML experiments showed effectiveness against AML cells and significantly increased mean overall survival.

**Effect of UCART123 on normal hematopoietic cells in vivo.** To evaluate the effect of UCART123 against normal hematopoietic

cells we used humanized xenografts (Hu- NSG) with either CD34 + CB cells ($N = 2$) or lymphocyte depleted BM cells from healthy donors. We evaluated the effect of UCART123 and TCRαβ KO cells with 10 million of each for Hu-CB cohorts and 2.5 million of each for Hu-nBM cohort. Differences were observed within the different subpopulations, they appear to vary between the Hu-NSG and also from mouse to mouse (Supplementary Fig. 4a–f). For example, there was a significant difference observed in the total percent of human cells in the Hu-NSG from CB-I after treatment with UCART123 treatment when compared to controls (Supplementary Fig. 4a). However, there were no significant differences in the progenitor (CD34[+]) cells when comparing UCART123 treatment with either TCRαβ KO or PBS controls (Supplementary Fig. 4e). Consistent with the CFU data, we observed a decrease in myeloid cells (CD33[+]) after treatment with UCART123 compared to TCRαβ KO in 2 out of the 3 Hu-NSG (CB-I *** $p = 0.001$, and nBM * $p = 0.046$). Such differences cannot be attributed to the different basal levels of CD123 (Supplementary Fig. 4b). Due to the high variability among all experimental cohorts, we developed an assay that would allow us to evaluate the effect of UCART123 on leukemic and normal cells within the same animal. We established a xenograft containing both normal and leukemic cells (mixed human T-cell depleted bone marrow (nBM) and an AML primary sample) in NSG mice (competitive xenograft nBM/AML2; Fig. 4a) as described[20,29]. Four weeks after injection of nBM and AML, human chimerism was confirmed and mice were injected with PBS, UCART123, or TCRαβ KO T-cells (one million each). Figure 4b shows a representative example of the presence of both normal and leukemic cells in the xenografts, demonstrating that AML cells have higher intensity of CD123 expression when compared to normal hematopoietic cells (also see Supplementary 4b). PB was monitored at different time points throughout the experiment. We found that over time, when the animals were left untreated, AML outcompeted the normal cells (Fig. 4c). In contrast, at day 16 after treatment, it was evident that the UCART123-treated mice showed only normal cells in the PB (Fig. 4c). After 5 weeks of treatment, mice were killed, and the BM was evaluated (Fig. 4d). We found that leukemic cells were selectively eliminated by UCART123 and most of the normal BM human cells were spared (Fig. 4d). In UCART123-treated mice, we found, on average, close to a 2-fold decrease in CD33 + cells (Fig. 4d), while lymphoid lineages were not impacted (Fig. 4d).

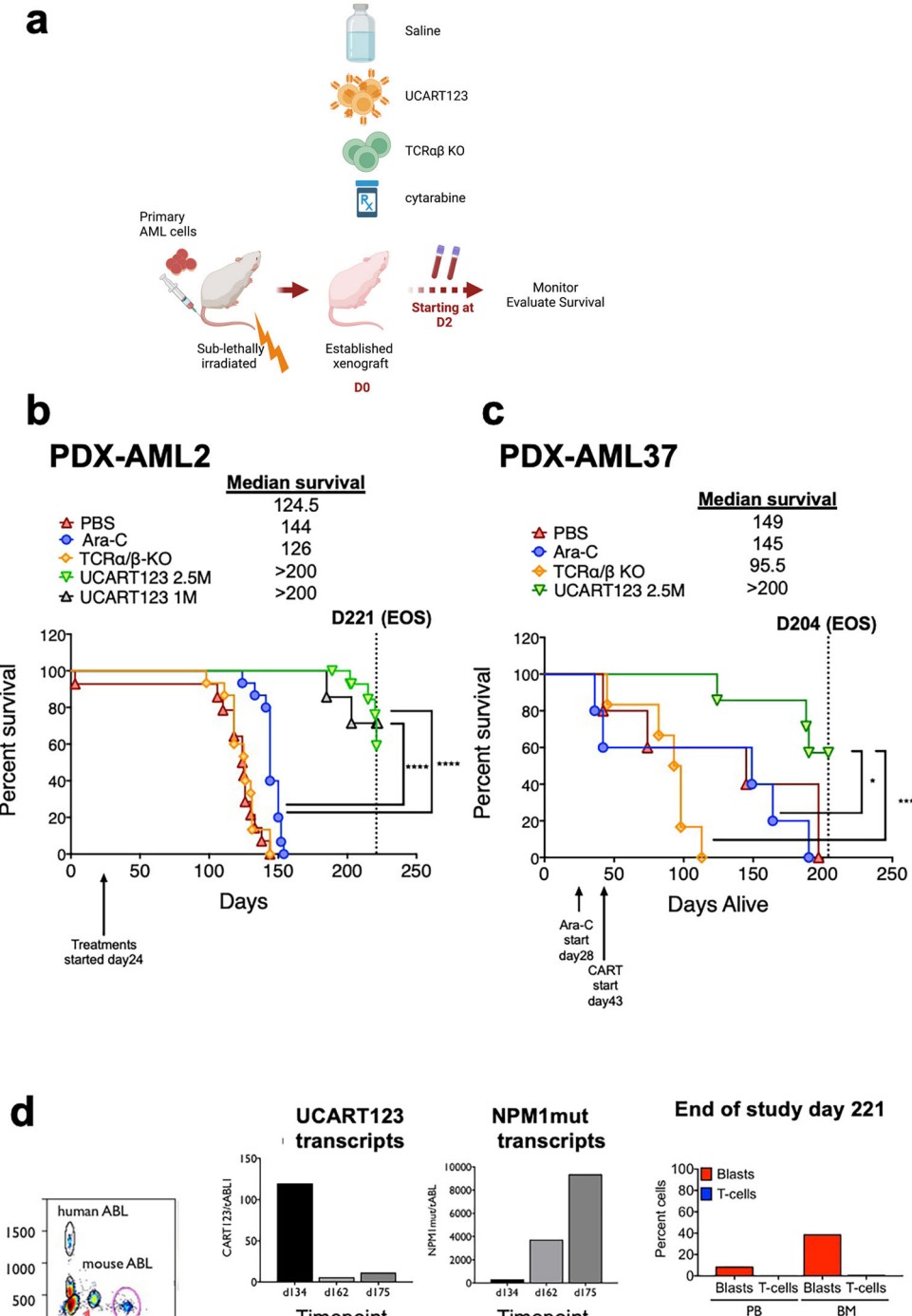

## Discussion

Here we present a universal CAR T cell approach using T cells from healthy donors. Key features of UCART123 include the TCRαβ knock out using TALEN® gene-editing technology to minimize the risk of GvHD; and, as safety feature, includes expression of the "safety switch" RQR8, allowing to eliminate UCART123 upon Rituximab treatment (Fig.1, Supplementary Fig. 1).

The specific anti-tumor activity of UCART123 against CD123 + primary AML samples was demonstrated in vitro using cytotoxicity and colony-forming assays. UCART123 induced a specific lysis of all AML samples tested. Moreover, we have shown an increased production of several cytokines such as IFN-γ, IL-2, IL-13, TNF-α, and IL-5 by UCART123 when cultured in the presence of all AML samples. Importantly, the inability of

**Fig. 3 UCART123 targets AML cells in vivo and results in improved overall survival in patient-derived xenografts (PDX) model. a** Schematic representation of the experimental procedure for the AML- PDX models (AML2 and AML37), showing that animals were treated with PBS, Ara-C, TCRαβ KO T cells or UCART123. Leukemia and T cells in peripheral blood (PB) or bone marrow (BM) were monitored starting on day 2 and later every 2–3 weeks and overall survival was evaluated. **b**, **c** Survival curves of PDX-AML2 cohorts treated with $1 \times 10^6$ UCART123 ($n = 7$), $2.5 \times 10^6$ UCART123 ($n = 15$) TCRαβ KO ($n = 15$), Ara-C ($n = 15$) or PBS (control; $n = 14$), and of PDX-AML37 cohorts treated with $2.5 \times 10^6$ UCART123 ($n = 7$) TCRαβ KO ($n = 6$), Ara-C ($n = 5$), or PBS (control; $n = 5$). Treatment with UCART123 significantly prolonged overall survival in PDX-AML2 (**b**) and PDX-AML37(**c**) compared to those in control, Ara-C, or TCRαβ KO. UCART123 $1 \times 10^6$ or $2.5 \times 10^6$ vs Ara-C; **** $p < 0.0001$, UCART123 $2.5 \times 10^6$ vs TCRαβ KO; *** $p = 0.0002$, UCART123 $2.5 \times 10^6$ vs Ara-C; * $p = 0.0171$, log-rank test. **d** NPM1 mutant in AML cells and UCART123 transcripts in PB were monitored by ddPCR on day 2 and every 2–3 weeks until mice are dead or killed due to sickness. The ddPCR plot shows simultaneous detection of human ABL1, mouse ABL1, NPM1 mutant, and UCART123 transcripts (left column). Representative examples are shown for a mouse that relapse (top) and one mouse that remained disease free (bottom). Evaluation of leukemia blasts and UCART123 cells in PB and BM at end of study on day 221 by flow cytometry demonstrated one mouse with progression of disease due to loss of UCART123 (top right) and another mouse in remission with sustaining UCART123 (bottom right). Source data are provided as a Source Data file.

primary leukemia samples to form colony units after treatment with UCART123 suggests their ability to eliminate leukemic progenitor cells.

CD123 has been reported to be highly expressed in most AML patients in all leukemic subpopulations: blasts, leukemic progenitor, and leukemia stem cells. Since CD123 can be expressed in normal hematopoietic progenitor and myeloid cells, we evaluated the potential toxicity of UCART123 on normal hematopoietic cells using two assays: cytotoxicity and colony-forming assays. We demonstrated a dose-dependent effect of UCART123 on normal myeloid progenitors as they showed a reduced myeloid colony formation only at the highest E:T ratios of UCART123 tested (5:1). However, at lower E:T ratios myeloid progenitors were not significantly affected and show to be capable of forming colonies in secondary CFU assays. Importantly, significant levels of cytotoxicity were observed with primary AML samples at these lower E:T ratios as well as lower ratios. These results demonstrate that there is increased toxicity of UCART123 against AML primary cells, as compared to the activity against myeloid progenitors.

The selective activity of UCART123 was also evaluated in vivo, using PDX-AML mice, and NSG mice humanized with human CD34[+] cells from CB or lymphocyte depleted BM. We demonstrated that mice bearing human AML (PDX-AML) treated with UCART123 presented significantly higher overall survival. In order to monitor residual disease and UCART cell persistence, we developed a ddPCR assay to simultaneously assess both cell populations, using NPM1-mutant AML as a proof-of-principle example. Using this assay, we demonstrated that persistently detectable UCART123 are capable of maintaining the animals without disease, even when re-challenged with AML cells.

Furthermore, despite the observed variability from sample to sample in the humanized mice (CB-I, CB-II, and nBM), we found that the expression of CD123 is lower in normal human hematopoietic cells when compared to AML blasts as has been reported in other studies[15]. UCART123 did not cause severe myeloablation in any of the studies (Supplementary Fig. 4c) as observed in other studies[26]. This observation is consistent with the preliminary results from a clinical trial with autologous CART against CD123 at the City of Hope where it was reported a lack of BM or any other organ toxicity[30]. Importantly, evaluation of CD34[+] progenitor cells after exposure to UCART123 suggested that hematopoietic progenitor cells were not ablated by UCART123 in the humanized mouse models (Supplementary Fig. 4e). However, as we began to utilize UCART123 in a clinical trial of patients with AML, effects on myeloid progenitor cells will be closely monitored, with the FDA-mandated requirements for donor availability and appropriate safety considerations for allogenic stem cell transplant if prolonged myelosuppression occurs. Because in the above experiments the activity of UCART123 was evaluated in the absence of leukemic

cells, we established a model in which leukemic cells could compete with normal hematopoietic cells to simulate what happens in vivo in patients with leukemia. We found that, in the competitive nBM/AML model, UCART123 preferentially targeted AML cells, with elimination of AML cells evident by day 16 after treatment. Such specificity is likely due to the difference in the overall intensity of expression of CD123 when compared to normal as we and others have reported[13,14,31–34] and also observed in the competitive nBM/AML model (Supplementary Fig. 4e). Analysis of CD34 + and CD33 + cells in the BM indicated that, while there was variability between individual mice, there was on average an approximately 2-fold decrease in CD33 + cells in the UCART123-treated mice but no decrease in the CD34 + cells, compared to the control PBS and TCRαβ KO treated mice. Taken together, the data suggest that the risk of ablation of hematopoietic stem cells in patients treated with UCART123 is limited, as UCART123 demonstrated preferential targeting of AML cells. Our experiments show that UCART123 effectively eliminates AML in vitro and in vivo without major impact against normal cells. A phase 1 trial of UCART123 in patients with relapsed/refractory AML is ongoing and results are eagerly anticipated.

## Methods

**Cells and cell culture**. Primary AML and BM specimens were obtained with informed consent under Weill Cornell Medicine IRB approval. CB and PBMCs from healthy donors were purchased from the New York Blood Center. Cell lines were obtained from the American Type Culture Collection (ATCC) or DSMZ-German Culture Collection of Microorganisms and Cell Cultures, all cell lines were authenticated and tested for mycoplasma. Mycoplasma contamination is monitored by using a colorimetric detection assay (PlasmoTest™, Mycoplasma Detection Kit, rep-pt1, InvivoGen) which is a cellular method based on the activation of Toll-Like Receptor 2. In addition, we confirm the results by using the LookOut® Mycoplasma PCR Detection Kit (MP0035, Sigma-Aldrich) optimized for use with JumpStart™ Taq DNA Polymerase (D9307, Sigma-Aldrich). Primary cryopreserved AML specimens were thawed and cultured for 1 h at 37 °C followed by treatment as described previously[35]. For primary sample characteristics see Supplementary Table 1. Cell lines were cultured in Iscove's Modified Dulbecco's Medium (IMDM; Life technologies) supplemented with 10–20% fetal bovine serum (FBS) according to culture conditions indicated by ATCC and 1% penicillin/streptomycin (Pen/Strep; Life Technologies).

**UCART123 cell production**. R&D grade UCART123 cells were produced by Cellectis using a large-scale manufacturing process, and non-transduced TCRαβ-deficient T-cells (TCRαβ KO T-cells) were produced from the same donor and in the same conditions, to be used as control cells.

Briefly, PBMCs from healthy donors were thawed and T-cells were activated using CD3/CD28 magnetic beads the day after thawing. Cultures were performed at 37 °C in 5% $CO_2$ in X-vivo15 media, supplemented with 5% Human AB Serum and 20 ng/ml of human IL-2. Cells were transduced with a recombinant lentiviral vector encoding the CD123-targeting CAR, at MOI 5, three days after activation. Two days later the cells were transfected with TALEN® mRNA (using Cellectis' proprietary CytoLVT-S electroporation system) to knockout the TRAC gene and disrupt TCRαβ expression.

Cells were then expanded at 37 °C in 5% $CO_2$ within a WAVE bioreactor Cellbag of 2 L or 10 L on the Xuri™ Cell Expansion System W25 for 10 days. At the

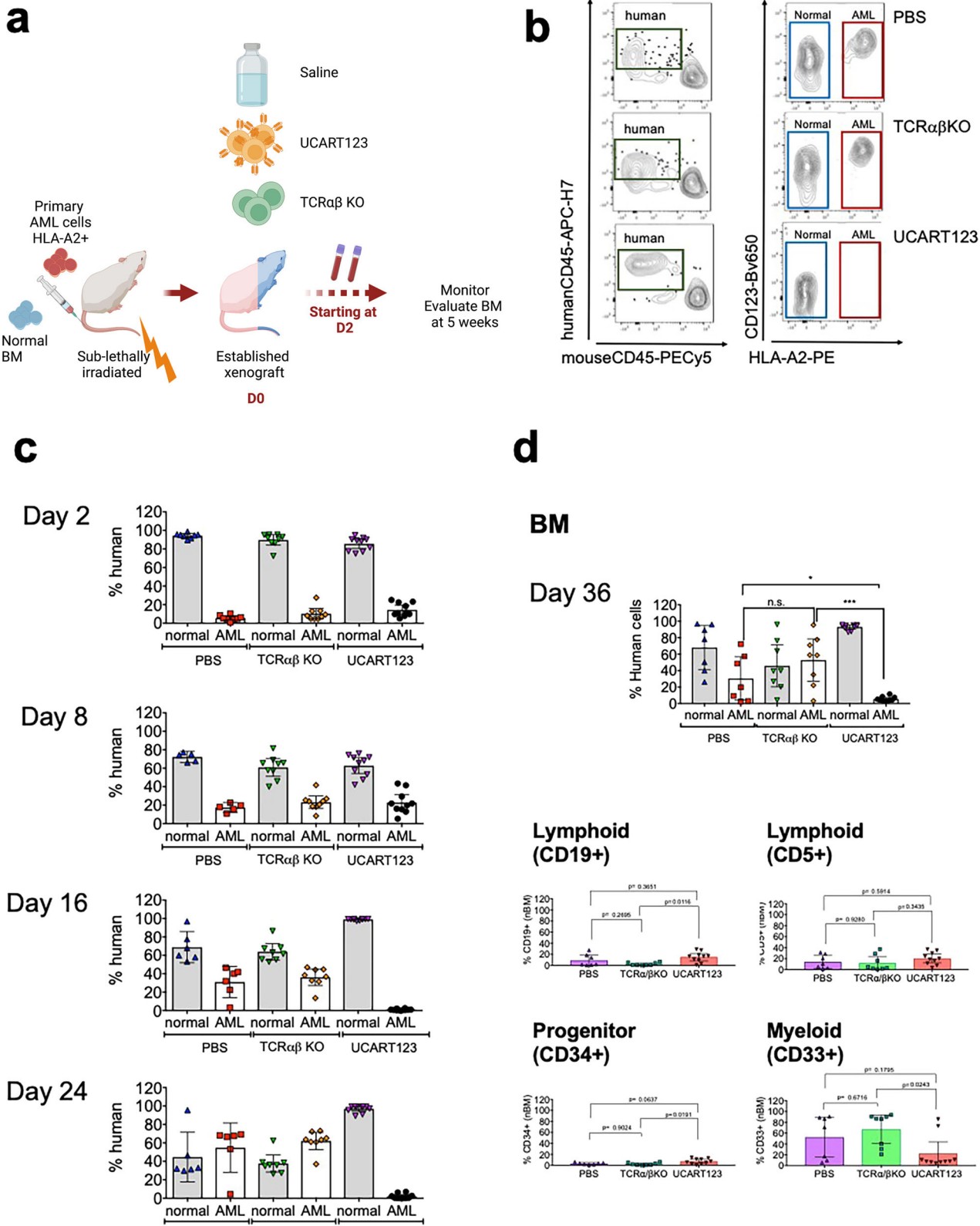

end of the expansion phase TCRαβ negative cells were isolated by negative selection using a TCRαβ biotin and anti-biotin magnetic bead system from Miltenyi, using the CliniMACS device. The purified TCRαβ(-) cell population was then resuspended in cryopreservation media, aliquoted in cryovials and stored at below -135 °C. The percentage of cells expressing the CAR targeting CD123 was 90% for the two UCART123 batches used in this study. For cell characteristics see Supplementary Table 2.

**Cell viability assay**. UCART123 cells were co-cultured at different E:T ratios (5:1, 1:1, and 0.5:1) with the different target cell populations (AML, CB, BM, or AML + CB co-cultures). When CB or AML cells were tested individually with UCART123, the target cells were stained with CFSE. For competitive cytotoxicity assays between CB and AML cells, CB and AML cells were stained with CFSE and CTV respectively when co-cultured together. After co-culturing primary AML cells and cell lines with UCART123 for 24 h, cell viabilities were measured by

**Fig. 4 UCART123 killed primary AML selectively in co-engrafted model with human normal hematopoiesis and human AML. a** Schematic representation for the competitive BM and AML model. Normal CD34[+] bone marrow cells (nBM) ($6 \times 10^6$ cells, HLA-A2-) were co-injected with T-cell depleted human AML cells ($2.5 \times 10^5$ cells, HLA-A2[+]) into sub-lethally irradiated NSG mice. After 7 weeks, mice were treated with PBS, $1 \times 10^6$ UCART123 or $1 \times 10^6$ TCRαβ KO T-cells. Normal hematopoietic cells and leukemia cells in PB and BM were monitored with flow cytometry starting on day 2 and later every 1–2 weeks. **b** Human cells gated with mouse CD45 versus human CD45 (left) were evaluated for CD123 expression level after normal cells (HLA-A2[−], blue rectangle) and leukemic cells (HLA-A2[+], red rectangle) were distinguished (right). Leukemic cells express higher CD123 levels than normal cells and were preferentially eliminated by UCART123 (right bottom). **c** Normal cells and leukemic cells were tracked in PB by flow cytometry starting on day 2 and at indicated timepoints. Leukemic cells outcompeted normal cells over time in PBS and TCRαβ KO cohorts; cohort size: $n = 5$ PBS, $n = 8$ TCRαβ KO, $n = 10$ UCART123. **d** Normal cells and leukemic cells were evaluated in BM on day 36 post treatment. PBS ($n = 5$) and TCRαβ KO ($n = 8$) cohorts showed progression of leukemia, while UCART123 ($n = 10$) treated mice did not (top). Evaluation of subsets in BM showed 2-fold decrease in CD33[+] myeloid cells in UCART123-treated mice, while lymphoid lineages were not impacted. Each symbol represents a mouse and bar represents the mean with the SD. \*\*\*$p < 0.001$, \*$p < 0.05$, one-way ANOVA. Source data are provided as a Source Data file.

multiparameter flow cytometry (MFC) of a BD-LSR II™ (BD) or a BD-LSRFortessa™ (BD) by staining with annexin V (BD Biosciences) and 7-aminoactinomycin (7-AAD, Life Technologies). For primary AML samples, cells were stained with CD34, CD123, and CD45 (BD Biosciences) and other surface markers (For a complete list of antibodies used see Supplementary Table 4) prior to annexin V/7-AAD staining. At least $2 \times 10^4$ events for cell lines or $2 \times 10^5$ events for primary AML samples were recorded per condition. Data were analyzed with FlowJo software (BD). Cells that were negative for annexin V and 7-AAD were scored as viable, and the viability of treated cells was normalized to untreated controls.

**Cytokine secretion assay**. The levels of INF-γ and other cytokines (IL-2, IL-4, IL-5, IL-6, IL-9, IL-10, IL-13, IL-17A, IL-17F, IL-21, IL-22, and TNF-α) were evaluated in supernatants obtained from 1:1 (E:T) ratio co-cultures after 24 h using the LEGENDplex™ Human Th Cytokine Panel (BioLegend).

**Colony-forming assay**. Colony-forming assay was performed as described[36,22]. Briefly, primary AML or cord blood cells were co-cultured with T-cells for 4 h. Cells were plated at $2.5 \times 10^4$ cells/ml in methylcellulose-based medium containing rh SCF, rh GM-CSF, rh IL-3 and rh EPO (MethoCult™ H4434 Classic (STEM CELL Technologies). Colonies were scored after 14 days of culture.

**Xenotransplantation assays with AML cell line**. Animal studies were performed under WCM IACUC approval. Non-obese diabetic/severe combined immunodeficient IL2Rgamma[null] (NSG) mice were obtained from the Jackson Laboratories. MOLM13 cells ($2.5 \times 10^5$ cell per mouse), AML cell line, expressing GFP and luciferase were injected into mice. Xenotransplanted mice were treated with different cell doses of UCART123 cells (expressed in total T-cells) starting on day 7. Leukemia burden was monitored weekly by bioluminescent imaging (BLI, In Vivo Preclinical Imaging Solutions (IVIS®), PerkinElmer) and overall survival was evaluated. For evaluation of rituximab (RTX) as a safety switch in this animal model, engrafted mice were treated with PBS (control) ($N = 8$), 10 million of UCART123 cells (UCART123 + vehicle) ($N = 12$) or 10 million of UCART123 cells followed by rituximab treatment (UCART123 + RTX) ($N = 12$) starting on day 7 post leukemia injection. Dose and schedule of RTX was 10 mg/kg/dose on days 2-6, 9-13, and 31-35 post UCART123 treatment. Surviving mice in groups UCART123/vehicle and UCART123/RTX and new control 4 NSG mice were re-challenged with additional injection of 1 million MOLM13 cells on day 43 post treatment to evaluate persistence of UCART123 cells. Leukemia burden was monitored weekly with BLI and UCART123 cells in PB or bone marrow aspirates were tracked with flow cytometry. Animals were killed when signs of distress were observed as required by WCM IACUC committee.

**Patient-Derived Xenotransplantation assays**. Animal studies were performed under WCM IACUC approval. AML patient-derived xenotransplants (AML-PDX) were first generated as described[37]. AML cells ($2.5 \times 10^6$ cells per mouse) were injected into sub-lethally irradiated NSG mice (270 rad). Peripheral blood (PB) was monitored and at the end of the experiment, bone marrow (BM) cells were harvested. Animals were killed when signs of distress were observed as required by WCM IACUC committee. Cells were stained with mouse CD45 (mCD45), human CD45 (hCD45) antibodies to identify human cells, additional antibodies were added to determine whether the human cells were AML or UCART123. For a complete list of antibodies used see Supplementary Table 4.

**Human normal BM (nBM)/AML competitive mouse assay**. Lymphocytes were depleted from BM and primary AML samples using CD3 and CD19 magnetic beads (Miltenyi). 6 million nBM or 6 million nBM with 0.25 million primary AML were injected into sublethally irradiated NSG mice for humanized mouse or competitive mouse assay respectively[29]. After 7 weeks, animals were started with the treatment. HLA-A2 antibody was added to distinguish nBM (HLA-A2 negative) and AML (HLA-A2 positive) to mouse CD45 (mCD45), human CD45

(hCD45) antibodies and other surface marker antibodies for evaluation by flow cytometry. For a complete list of antibodies used see Supplementary Table 4.

**Droplet digital PCR**. In PDX models, UCART123 transcripts in peripheral blood (PB) and bone marrow (BM) were evaluated with ddPCR at day 2, day 14, at days of follow-up and at the timepoint of euthanasia due to sickness or disease progression. RNA was extracted from cells harvested from PB and BM by using AllPrep RNA/DNA kit (Qiagen) and 25-50 ng of tRNA was converted into cDNA with SuperScript VILO (Thermo Fisher Scientific). Ten million droplets were generated from cDNA samples (step 1) and single molecule in droplets were amplified by PCR reaction (step2) followed by counting the absolute number of fluorescent droplets (step 3). Step 1 and step 3 were performed and analyzed with RainDrop and RainDrop Analysist II software (RainDance Technologies/Bio-Rad Technologies). Copy numbers of UCART123 transcript relative to $1.0 \times 10^4$ ABL1 transcript (reference) were calculated. Primers provided in Supplementary Table 5.

**Reporting summary**. Further information on research design is available in the Nature Research Reporting Summary linked to this article.

## Data availability

The data that support the findings of this study are provided within the Article, Supplementary Information or Source Data file. Source data are provided with this paper.

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

## Acknowledgements
MLG receives support from the following awards R21AG066552, R21CA245454, R01 CA234478 and Evans Foundation Discovery Grant.

## Author contributions
Experiments and analysis were performed by M.S., R.G., H.Z., N.E.C., V.T.A., N.M.T., W.Y., S.F., C.L., and A.G., supervised by J.S., D.C.H., G.J.R., and M.L.G. The manuscript was written by M.S., R.G., A.G., J.S., G.J.R., and M.L.G., and all authors read and provided feedback.

## Competing interests
The authors declare the following competing interests: Research funding from Cellectis (M.S., H.Z., N.E.C., V.T.A., N.M.T., W.Y., D.C.H., M.L.G., and G.J.R.). Employees from Cellectis (R.G., S.F., C.L., A.G., and J.S.).
