## [Peer Review File · Nature Communications]

Reviewers' Comments:

Reviewer #2:

Remarks to the Author:

Sugita et al generated allogeneic TCRab deficient CAR T cells targeting CD123, which is upregulated in AML, and showed UCART123 effectively inhibited human AML in vitro and in vivo in several models including AML PDX. They also showed UCART123 T cells have relatively low toxicity to human hematopoietic progenitors in vitro and in BM/AML PDX-co-engraftment experiments. Overall, the work is important, and "universal" CAR T cells, which are supposed to have no or low grafts versus host disease (GvHD), will be quite valuable for expanding the use of the treatment as one batch of the CART production can be used for multiple patients. Moreover, UCART123 appeared to have low toxicity to the HSCs, and it is interesting the bone marrow/AML PDX co-engraftment was used. However, the analysis of the potential persistence of the HSC function was too preliminary at it is. There are some major concerns about the studies. First, one major advantage of the investigation is to generate and evaluate "universal" CART, yet not attempt was made or experimental studies were designed to address the intended "universal" CART function, or reduction/elimination of GvHD. Second, while the low toxicity of UCART123 to human progenitor cells was impressive, the analysis of the HSCs parameters was way preliminary, and inconsistent with about 60% reduction in CD33+ cells from the UCART123 treated mice. Third, TALEN technology is relatively old way to generate KO of a gene, and it is unclear why more efficient CRISPR method was not used to generate the TCRa/b KO T cells. There are some more detailed points the authors need to address as below:

1. Figure 1. It is unclear why normal untransduced T cells were not used in this studies, as normal T cells, even not CARTs, could have certain background activity attacking some tumor cells. Please also provide the TCRa KO efficiency and test the expression of TCRa/b on the T cell surface.
2. Figure 2. As the mice were kept until >80 days following the CART injection, TCR intact CART123 could be used to address whether UCART123 has reduced or lost capacity to induce GvHD. If CART123 is used as control, will the authors observe the enhanced T cell proliferation in UCART123 treated mice in comparison with the CART treated mice? Were TCR repertoire different and changed in UCART123 as compared to CART123? This should be addressed in Discussion.
3. Figure 4. Some groups reported CD123 CAR T cells have strong toxicity to human bone marrow and HSCs in HIS mouse model, leading to myeloablation (Gill S. et al, Blood, 2014; Tassian, SK et al, Blood, 2017). Why the UCART123 results are so different from the other reported toxicity to HSC function? Was different scFv for different epitope used? Or there are differential effect for BM from different donors, including the age difference or other factors?
4. Figure 4. The authors used normal bone marrow cells ~24 times the number of T-cell depleted AML cells in the mixed xenograft model, why would they choose this ratio?
5. Figure 4D: More detailed and rigid analysis of the impact of UCART123 should be performed. For instant, what is the total number of HSC-enriched population in BM and spleens, in particular, the CD34+/CD38-/Lin- population in each group? What is the percentage of this population among the total human CD45+/HLA-A2- cells?
6. Figure 4D. It is not clear why CD33+ cell reduced two fold. It is valuable to assess, in both or either BM and spleens, the profile of lineages, such as HLA-A2-CD3+ cells CD19/20+, and CD33+, to gain a glimpse of the over lineage distribution of the control and UCART123 treated mice. Is it likely that UCART123 skewed lineage profile by attacking CD33+/CD123+ mature or differentiated cells yet spare HSCs and thus human HLA-A2- T and B cell lineages?
7. Minor: the supplementary data start with "extended data figure 2".

Reviewer #3:

Remarks to the Author:

The manuscript by Sugita et al. highlights the activity of UCART123 cells, T cells genetically modified to express CD123 and gene edited to knock out TCR a/b using TALEN technology. Knocking out TCR a/b allows the product to be used off-the shelf and minimizing risk of GvHD. In addition, they posit that UCAR123 T cells do not have significant myelotoxicity. They show antitumor activity of UCAR123 T cells against MOLM13 cell in vitro and in vivo. In vivo, treating with UCAR123T cells at different doses translates to increased survival advantage in those groups. This is then also shown in 2 different PDX models. Determination of myelotoxicity by UCAR123 T cells was assessed by a flow cytometry-based cytotoxicity assay, CFU assays, as well as in vivo by xenografting BM and AML cells. The use of gene editing to knock out TCR a/b has been previously described in CD19 CAR T cells. Targeting CD123 using a CAR construct, albeit not in a UCART, has also been pursued by other groups.

Some aspects that would have made the manuscript stronger:

- Describing the CAR construct in detail would be helpful (e.g. second generation, which scFv? Costimulatory domain only mentioned in the figure but not alluded to in the text). Given that the main premise of novelty here is the UCART platform, it would be valuable to spend more time describing the use of TALEN to KO TCR a/b.
- The experiments compare UCAR123 T cells to control TCRa/b KO T cells, but the comparison to normal T cells expressing CD123 CAR is not performed. Showing that UCAR123 are at least as effective as their non gene edited counterpart would be desirable.
- In vitro studies using MOLM13 would be strengthened by using additional cell lines (e.g. THP-1, MV4-11, etc.)
- The animal experiments use different T cell doses each time, and this is not justified or explained in the discussion. In one instance, mice engrafted with CB receiving 10×10^6 and mice engrafted with BM receiving 2.5×10^6 , samples were then mixed and analyzed together, introducing an additional confounding variable.
- Extended figure 1 is missing
- Statistical analysis is lacking in some experiments, with notable absence of some N and p values. (e.g. Figure 1 b,c)

The flow of the manuscript can be improved by rearranging the order in which the experiments are described. In addition, the discussion would benefit from evaluating the manuscript findings and comparing to what has been described in the CD123 CAR literature.

Reviewer #4:

Remarks to the Author:

In the present manuscript the authors present pre-clinical data for UCART123, an off the shelf CAR-T product already explored in the clinic. The manuscript is very well and clear written. The experiments seem to be adequate performed and are well analyzed. Some minor comments are provided further below. Although the overall data set on the pre-clinical evaluation of UCART123 provided in the manuscript is impressive, novelty is lacking as many groups have published pre-clinical data on CD123 CAR-T targeting AML (e.g. Arcangeli et al. 2017, Pero et al. 2018). The only novel aspect is that the authors used an off-the-shelf approach. To strengthen the novelty of their approach, the authors should more specifically highlight the differentiators of their approach to already published autologous approaches. How many patients could be treated with a single batch UCART123? How critical is timing in the clinical setting that off the shelf would indeed offer an advantage to an autologous product? Furthermore, the authors claim from their data that only modest toxicity was observed against the stem cell compartment, very likely depending on the target density. From the presented data it remains unclear how this would be translated into the clinical setting and how long-term toxic effects against the myeloid compartment could be mitigated? Does the approach require ablation of the UCART123 after they have done their job? If so feasibility of UCART123 elimination needs to be demonstrated by e.g. adding rituximab. Furthermore, mitigation of acute toxicities is not addressed. This aspect is especially of relevance as UCART123 related deaths were reported from clinical studies. The authors should also discuss advantages/disadvantages of UCART123 in comparison to so called switchable CAR-T approaches which could help to avoid CD123-related toxicities at all (e.g. Loff et al. 2020).

In summary the reviewer would recommend the manuscript for publication if these aspects are

addressed.

Some more detailed comments are made in the following:

The authors found a reduced cytotoxic response against normal hematopoietic precursor cells derived from CB in comparison to AML samples. It would be very helpful to interpret the data if the target antigen density on AML and CB cells would be shown. Furthermore, that authors performed CFA after treatment and observed a reduction of CFU resulting from UCART123 treated AML samples compared to CB samples. The authors concluded that these results indicate a reduction of LICs in the AML samples. This effect could be also due to a bystander effect on normal hematopoietic cells in the sample by AML-directed UCART123 effects (e.g. cytokines)? The authors might draw their conclusion more carefully.

Figure 1B – in vitro killing assay against MOLM13 seems to result in a max. killing plateau at ~ 60% dead cells at higher e:t ratios despite the fact that MOLM13 are CD123 high. Is this a technical artefact or a limitation of UCART123 killing abilities?

Figure 2A – color code to the different AML samples is missing; Figure 2C shows high standard deviation – explanation? From the legend it is unclear if the graph shows a single CB donor with 6 replicative or several donors? In Figure 2D there are no negative values in comparison to 2C for CB – explanation? Only a single AML sample is used for comparison – how does CD123 density on this AML sample compares to other samples?

Figure 3C – treatment with UCART123 was started later than in 3B – why? Why ARA-C treatment was started earlier? Number of animals per group is missing.

Figure 4 – which AML sample was used for the experiment? How does antigen density compare to other AML samples?

Figure 4C – there seems to be an effect on the normal hematopoietic compartment in the presence of UCART123 – please provide explanation.

NCOMMS-20-29505-T “Allogeneic TCR $\alpha\beta$ Deficient CAR T-Cells Targeting CD123 in Acute Myeloid Leukemia (AML)”

We thank the reviewers for taking the time to carefully and critically review our manuscript. We feel we have addressed each of the concerns. In addition, all references have also been updated. Please find below point by point our responses shown in red. We believe that the manuscript is more clear and stronger after incorporating and addressing all the concerns of the Reviewers.

REVIEWER COMMENTS

Reviewer #2 (Remarks to the Author):

Sugita et al generated allogeneic TCR $\alpha\beta$ deficient CAR T cells targeting CD123, which is upregulated in AML, and showed UCART123 effectively inhibited human AML in vitro and in vivo in several models including AML PDX. They also showed UCART123 T cells have relatively low toxicity to human hematopoietic progenitors in vitro and in BM/AML PDX-co-graftment experiments. Overall, the work is important, and “universal” CAR T cells, which are supposed to have no or low grafts versus host disease (GvHD), will be quite valuable for expanding the use of the treatment as one batch of the CART production can be used for multiple patients. Moreover, UCART123 appeared to have low toxicity to the HSCs, and it is interesting the bone marrow/AML PDX co-graftment was used. However, the analysis of the potential persistence of the HSC function was too preliminary at it is.

There are some major concerns about the studies.

First, one major advantage of the investigation is to generate and evaluate “universal” CART, yet not attempt was made or experimental studies were designed to address the intended “universal” CART function, or reduction/elimination of GvHD.

We thank the reviewer for the comment, we have now added data that shows the differences between UCART123 (CAR+ TCR $\alpha\beta$ -) compared to unmodified T-cells (or not transduced NTC). **Extended figure 1c** shows the body weight curves of NSG mice comparing the UCART123 cells (99.8% TCR $\alpha\beta$ -) and unmodified T cells (NTD) from the same donor (8.9% TCR $\alpha\beta$ -). All mice injected with TCR $\alpha\beta$ -positive unmodified T-cells showed treatment-related changes consistent with GvHD (weight loss and histopathological changes). In addition, in **Extended figure 1d** we show representative figures for the histopathological changes observed in mice injected with NTC cells. Conversely, UCART123 did not elicit any sign of GvHD in the selected model and during a follow-up period of up to 80 days (including histopathology analysis of selected organs). These results give thus confidence that UCART123 is unlikely to induce a severe GvHD when administrated to patients.

In addition, UCART19, a product with similar features (same TALEN-induced TRAC knock-out) has not shown severe GVHD in clinical studies (**Benjamin, R et al., Lancet 2020 396:1885-94**).

Second, while the low toxicity of UCART123 to human progenitor cells was impressive, the analysis of the HSCs parameters was way preliminary, and inconsistent with about 60% reduction in CD33+ cells from the UCART123 treated mice.

To clarify this point to the reviewer, we do see some toxicity to normal myeloid progenitors (as expected) and such toxicity is dose dependent (see **figure 2f**) this data is consistent with the in vivo reduction in myeloid cells. To clarify the decrease toxicity to stem/progenitor cells, we are now showing secondary colony assays from co-cultures at 1:1 ratios (**Extended figure 2b**), the data shows non-significant changes in the ability of the cells to generate myeloid progeny, and suggesting that HPSCs have not been significantly affected by UCART123 treatment.

Secondary plating (1:1)

Third, TALEN technology is relatively old way to generate KO of a gene, and it is unclear why more efficient CRISPR method was not used to generate the TCRa/b KO T cells.

We would like to respectfully point out that we were able to achieve an efficient KO of the TRAC gene (>90%) using TALEN gene editing technology. Furthermore, the objective of the paper is to provide preclinical proof of concept of allogeneic CAR T cells targeting CD123 for therapeutic use. Indeed, the main advantage of TALEN is their target recognition system, allowing them to be extremely precise, specific to their target and efficacious, due in part to the length of their 30+ base pairs binding site. Therefore this specificity makes TALEN a preferred technology for therapeutic applications.

There are some more detailed points the authors need to address as below:

1. Figure 1. It is unclear why normal untransduced T cells were not used in this studies, as normal T cells, even not CARTs, could have certain background activity attacking some tumor cells. Please also provide the TCRa KO efficiency and test the expression of TCRa/b on the T cell surface.

We performed the initial experiments using non-transduced T cells (NTC) as a control as shown in **extended data figure 1**, and the NTC cells result in GVHD. We consider that for this study (since the costs of each in vivo experiment is high) the most appropriate comparison was the

TCRab KO cells from the same donor. In addition, by performing long-term in vivo evaluation with TCRab KO we avoid animal death due to GvHD or other allogeneic T-cell effect on tumor cells. As suggested by the reviewer, now we are including in **extended data figure 1b** a representative flow cytometry example for the TCR expression on the cells used for the study, in addition to a table for all the batches tested in **supplemental table 2**.

Furthermore, the data comparing CART123 cells, NTC cells and UCART123 cells is shown below. These data are shown in the manuscript submitted simultaneously (testing UCART123 in BPDCN) by our collaborator Dr. Konopleva. Specifically, the data presented below shows the in vitro activity of T-cells expressing CAR123 against different CD123 expressing cell lines. Daudi cells were used as a CD123 negative control, while KG1a, MOLM13 and RPMI-8226 were used as targets expressing increasing levels of CD123. The left panel shows IFNγ levels released in the supernatants of overnight co-cultures of CAR123 cells with target cells at 1:1 E:T ratio, using an ELISA test. Cytotoxic activity against MOLM13 and RPMI-8226 cells is shown in the right panel. Co-cultures were settled with each of the CD123+ cell lines (together with Daudi CD123^{neg} cells as an internal negative control) at a 10:1 effector to target (E:T) ratio for 18 hours. Cytotoxic activity was evaluated by assessing the viability of the different target cell populations by flow cytometry at the end of the co-culture. Cell killing activity was normalized to the activity against CD123- Daudi cells.

NTD: non transduced T-cells. **UCART123:** T-cells transduced with CAR123, transfected with *TRAC* targeting TALEN[®] and purified by depletion of remaining TCRab⁺ T-cells. **CART123:** T-cells transduced with CAR123, mock transfected (TCRab⁺ cells). The experiments were performed with T-cells from three different donors, each of which is represented by a different color (black, green or white symbols). The donor highlighted in black was transduced with a R&D backbone, while the two other donors were transduced with an rLV produced from the same backbone used to manufacture UCART123 clinical batches. Nevertheless, all three donors express the CAR from the same EF1 α -RQR8-2A-CAR123 lentiviral expression cassette.

2. Figure 2. As the mice were kept until >80 days following the CART injection, TCR intact CART123 could be used to address whether UCART123 has reduced or lost capacity to induce GvHD. If CART123 is used as control, will the authors observe the enhanced T cell proliferation in UCART123 treated mice in comparison with the CART treated mice? Were TCR repertoire different and changed in UCART123 as compared to CART123? This should be addressed in Discussion.

Unfortunately we do not have stored material to test the TCR repertoire. The *in vivo* experiments are long and expensive to repeat. However, we are now including data showing that UCART123 cells do not induce GvHD unlike the TCR intact donor cells.

3. Figure 4. Some groups reported CD123 CAR T cells have strong toxicity to human bone marrow and HSCs in HIS mouse model, leading to myeloablation (Gill S. et al, Blood, 2014; Tassian, SK et al, Blood, 2017). Why the UCART123 results are so different from the other reported toxicity to HSC function? Was different scFv for different epitope used? Or there are differential effect for BM from different donors, including the age difference or other factors?

To specifically address this issue, we generated the competitive nBM/AML model. This model allows for the simultaneous evaluation of activity against normal and malignant cells. Similarly, we evaluated in vitro UCART123 toxicity in co-cultures of AML and CB cells (**Figure 2d**). Furthermore, when performing serial CFU assays, we only observed toxicity to normal myeloid progenitors when using higher numbers of UCART123 cells (**Figure 2f**). Finally, we observed that UCART123 cells are more likely to eliminate AML cells (that expressed high levels of CD123, as shown in **Figure 4b**) than normal cells, using the competitive assays. It is likely that by using lower doses of UCART123 we improved the therapeutic window ,.

Furthermore, it should be noted that in ongoing early stage clinical trials with autologous CART and anti-CD123 monoclonal antibodies the myelosuppression has not been observed. In addition, the FDA mandates availability of the potential allogenic stem cell transplant donor for any patient undergoing anti-CD123 CART therapy.

4. Figure 4. The authors used normal bone marrow cells ~24 times the number of T-cell depleted AML cells in the mixed xenograft model, why would they choose this ratio?

The ratios of nBM and AML cells were based on the prior experience of our laboratory working with the AML cells in prior transplantation experiments. For the nBM cells since they were only lymphodepleted we transplanted the expected number of cells needed for a successful engraftment of healthy HSPCs. This approach was based on our previous publication (Guzman ML, PNAS 2002).

5. Figure 4D: More detailed and rigid analysis of the impact of UCART123 should be performed. For instant, what is the total number of HSC-enriched population in BM and spleens, in particular, the CD34+/CD38-/Lin- population in each group? What is the percentage of this population among the total human CD45+/HLA-A2- cells?

We thank the reviewer for the suggestion, however, we were limited by the number of tubes we can evaluate by flow cytometry for a large number of events and the number of parameters we are able to assess to be able to distinguish, mouse and human cells, nBM and AML. Thus, we did not include a complete lineage cocktail or CD38 in the evaluation.

6. Figure 4D. It is not clear why CD33+ cell reduced two fold. It is valuable to assess, in both or either BM and spleens, the profile of lineages, such as HLA-A2-CD3+ cells CD19/20+, and CD33+, to gain a glimpse of the over lineage distribution of the control and UCART123 treated mice. Is it likely that UCART123 skewed lineage profile by attacking CD33+/CD123+ mature or differentiated cells yet spare HSCs and thus human HLA-A2- T and B cell lineages?

We thank the reviewer for the suggestion, as mentioned above we were limited by the number of tubes we can evaluate by flow cytometry for a large number of events and the number of parameters we are able to assess to be able to distinguish, mouse and human cells, nBM and AML. However, we re-analyzed the data separating CD33 and CD123 populations. We are showing below the different subpopulations for all the treatments. The data suggest that overall CD33+ populations are the most affected by the UCART123 treatment.

Figure legend: A competitive BM and AML model was established by co-injecting normal (lymphocyte depleted) bone marrow cells (nBM) (HLA-A2-) were co-injected with T-cell depleted human AML cells (HLA-A2+) into sub-lethally irradiated NSG mice. After 7 weeks, mice were treated with PBS, 1×10^6

UCART123 or 1×10^6 TCR $\alpha\beta$ KO T-cells. Normal hematopoietic cells and leukemia cells in PB and BM were monitored with flow cytometry starting on day 2 and later every 1-2 weeks. b, Human cells gated with mouse CD45 versus human CD45 and normal cells (HLA-A2-) were evaluated for CD33 and CD123 expression in BM on day 36 post treatment.

7. Minor: the supplementary data start with “extended data figure 2”.

We apologize for the oversight. This has been corrected.

Reviewer #3 (Remarks to the Author):

The manuscript by Sugita et al. highlights the activity of UCART123 cells, T cells genetically modified to express CD123 and gene edited to knock out TCR a/b using TALEN technology. Knocking out TCR a/b allows the product to be used off-the shelf and minimizing risk of GvHD. In addition, they posit that UCAR123 T cells do not have significant myelotoxicity.

They show antitumor activity of UCAR123 T cells against MOLM13 cell in vitro and in vivo. In vivo, treating with UCAR123T cells at different doses translates to increased survival advantage in those groups. This is then also shown in 2 different PDX models.

Determination of myelotoxicity by UCAR123 T cells was assessed by a flow cytometry-based cytotoxicity assay, CFU assays, as well as in vivo by xenografting BM and AML cells. The use of gene editing to knock out TCR a/b has been previously described in CD19 CAR T cells. Targeting CD123 using a CAR construct, albeit not in a UCART, has also been pursued by other groups.

Some aspects that would have made the manuscript stronger:

-Describing the CAR construct in detail would be helpful (e.g. second generation, which scFv? Costimulatory domain only mentioned in the figure but not alluded to in the text). Given that the main premise of novelty here is the UCART platform, it would be valuable to spend more time describing the use of TALEN to KO TCR a/b.

We thank the reviewer for the comment, we have now added more information on the construct used in the text of the manuscript as follows: The CAR construct selected for use in UCART123

combines the scFv derived from a murine hybridoma, a CD8 hinge and transmembrane domain, and a cytoplasmic tail composed of 4-1BB co-stimulatory domain and CD3zeta signaling domain. The lentiviral vector cassette that drives CAR expression has been designed to co-express RQR8 (through a 2A peptide linker) under the control of an EF1 α promoter (detailed in **Figure 1a**). RQR8 is an artificial cell surface protein combining antibody-binding epitopes from both human CD34 and human CD20 antigens (Philip et al., 2014). The CD20 epitopes present within the construct are recognized by rituximab, thus enabling deletion of RQR8-expressing cells through complement mediated cell killing (CDC) and antibody-dependent cell-mediated cytotoxicity (ADCC) in the presence of rituximab.

-The experiments compare UCART123 T cells to control TCR α /b KO T cells, but the comparison to normal T cells expressing CD123 CAR is not performed. Showing that UCART123 are at least as effective as their non gene edited counterpart would be desirable.

We thank the reviewer for the comment. We are co-submitting a manuscript with the same UCART123 construct for BPDCN with Dr. Konopleva, where we have addressed this concern by adding data that shows the differences between UCART123 (CAR $^{+}$ TCR $\alpha\beta^{-}$) compared to unmodified T-cells (or not transduced NTC). Dr Konopleva's manuscript contains the figure shown below, demonstrating that the TCR α /b deletion in UCART123 does not affect the activity of the CAR123 T cells (CART123) against CD123 $^{+}$ cells (IFN γ secretion and cytotoxicity shown).

Figure legend: In vitro activity of T-cells expressing CAR123 against cell lines expressing different levels of CD123. Daudi cells were used as a CD123 negative control, while KG1a, MOLM13 and RPMI-8226 were used as targets expressing increasing levels of CD123. **Left panel** shows IFN γ levels released in the supernatants of overnight co-cultures of CAR123 cells with target cells at 1:1 E:T ratio, using an ELISA test. Cytotoxic activity against MOLM13 and RPMI-8226 cells is shown in **Right panel**. Co-cultures were settled with each of the CD123+ cell lines (together with Daudi CD123^{neg} cells as an internal negative control) at a 10:1 effector to target (E:T) ratio for 18 hours. Cytotoxic activity was evaluated by assessing the viability of the different target cell populations by flow cytometry at the end of the co-culture. Cell killing activity was normalized to the activity against CD123- Daudi cells. **NTD:** non transduced T-cells. **UCART123:** T-cells transduced with CAR123, transfected with *TRAC* targeting TALEN[®] and purified by depletion of remaining TCR $\alpha\beta$ ⁺ T-cells. **CART123:** T-cells transduced with CAR123, mock transfected (TCR $\alpha\beta$ ⁺ cells). The experiments shown were performed with T-cells from three different donors, each of which is represented by a different color (black, green or white symbols). The donor highlighted in black was transduced with a R&D backbone, while the two other donors were transduced with an rLV produced from the same backbone used to manufacture UCART123 clinical batches. Nevertheless, all three donors express the CAR from the same EF1 α p-RQR8-2A-CAR123 lentiviral expression cassette.

-In vitro studies using MOLM13 would be strengthened by using additional cell lines (e.g. THP-1, MV4-11, etc.)

As suggested by the reviewer, we are including data for 4 additional CD123+ cell lines co-cultured with UCART123 at E:T ratios of 1:1 (refer to Extended Fig 1a shown below). All cell lines were susceptible to UCART123 killing. This data is consistent with our observations using primary AML samples.

-The animal experiments use different T cell doses each time, and this is not justified or explained in the discussion. In one instance, mice engrafted with CB receiving 10×10^6 and mice engrafted with BM receiving 2.5×10^6 , samples were then mixed and analyzed together, introducing an additional confounding variable.

We apologize for the lack of clarity. The initial experiments were started at doses most commonly used for CART cells, as it was also done with the cell line xenograft experiments (i.e., 10×10^6 cells per mouse). Since our in vitro data strongly suggested that UCART123 cells have potent activity at low E:T ratios, we tested lower doses of UCART123 (2.5×10^6 and 1×10^6). We found that doses as low as 1×10^6 per mouse show potent anti-leukemia activity.

-Extended figure 1 is missing

We apologize for the oversight. This has been corrected.

-Statistical analysis is lacking in some experiments, with notable absence of some N and p values. (e.g. Figure 1 b,c)

We apologize for the oversight. This has been corrected.

The flow of the manuscript can be improved by rearranging the order in which the experiments are described. In addition, the discussion would benefit from evaluating the manuscript findings and comparing to what has been described in the CD123 CAR literature.

We have added additional CAR literature and changed the manuscript for clarity.

Reviewer #4 (Remarks to the Author):

In the present manuscript the authors present pre-clinical data for UCART123, an off the shelf CAR-T product already explored in the clinic. The manuscript is very well and clear written. The experiments seem to be adequate performed and are well analyzed. Some minor comments are provided further below. Although the overall data set on the pre-clinical evaluation of UCART123 provided in the manuscript is impressive, novelty is lacking as many groups have published pre-clinical data on CD123 CAR-T targeting AML (e.g. Arcangeli et al. 2017, Pero et al. 2018). The only novel aspect is that the authors used an off-the-shelf approach. To strengthen the novelty of their approach, the authors should more specifically highlight the differentiators of their approach to already published autologous approaches. How many patients could be treated with a single batch UCART123? How critical is timing in the clinical setting that off the shelf would indeed offer an advantage to an autologous product? Furthermore, the authors claim from their data that only modest toxicity was observed against the stem cell compartment, very likely depending on the target density. From the presented data it remains unclear how this would be translated into the clinical setting and how long-term toxic effects against the myeloid compartment could be mitigated? Does the approach require ablation of the UCART123 after they have done their job? If so feasibility of UCART123 elimination needs to be demonstrated by e.g. adding rituximab. Furthermore, mitigation of acute toxicities is not addressed. This aspect is especially of relevance as UCART123 related deaths were reported from clinical studies. The authors should also discuss advantages/disadvantages of UCART123 in comparison to so called switchable CAR-T approaches which could help to avoid CD123-related toxicities at all (e.g. Loff et al. 2020).

We thank the reviewer for the thoughtful comments. We have expanded in the manuscript text on the advantages of an off-the-shelf product vs the autologous product. We will also expanded on our speculation regarding the preference for leukemia cells over normal stem/progenitor cells. In addition, since the cells are allogeneic, the recipient cells are likely to eliminate them once the lymphoid lineage is recovered. However, we are also including feasibility data for the use of rituximab to eliminate UCART123 cells in vivo. (**Extended data 1e**).

UCART123 cell can be eliminated by rituximab. MOLM13-BLIV bearing mice (NSG) were treated with Saline (N=3) or 3M UCART123 cells (N=14). Seven days after treatment the 14 mice were randomized into two groups, Saline (N=6) and Rituximab 10mg/kg (N=8) and were treated days 7-11 and days 37-41. Leukemia was monitored by radiance. Figure shows a representative example of the radiance measured at day 52 in the surviving UCART123 treated animals showing disease progression in the group treated with rituximab.

Some more detailed comments are made in the following:

The authors found a reduced cytotoxic response against normal hematopoietic precursor cells derived from CB in comparison to AML samples. It would be very helpful to interpret the data if the target antigen density on AML and CB cells would be shown.

We thank the reviewer for the comment, we have now cited our previous work (Cruz, et al., Leuk Lymphoma. 2018 Apr; 59(4): 978–982) where we evaluated the expression of CD123 across primary AML samples and healthy donors. Below, we are showing a representative example (left panel) for the mean fluorescence intensity (MFI) detected in primary AML samples compared to healthy donors. Further, this was validated in the competitive AML/nBM model where we show a differential expression of CD123 in the HLA-A2+ (AML) and the HLA-2A- (normal) cells (**Figure 4b-** also shown below right panel). Thus, as the reviewer points out, this

is the likely the reason for the decreased toxicity against normal cells (together with the use of a lower dose of UCART123 cells).

Furthermore, that authors performed CFA after treatment and observed a reduction of CFU resulting from UCART123 treated AML samples compared to CB samples. The authors concluded that these results indicate a reduction of LICs in the AML samples. This effect could be also due to a bystander effect on normal hematopoietic cells in the sample by AML-directed UCART123 effects (e.g. cytokines)? The authors might draw their conclusion more carefully.

We agree with the reviewer to tone down the conclusion from the colony forming assays as we did not formally test the leukemia initiating capacity of the cells. Importantly, we have clarified the text to point out that we performed the experiment in two different ways. One was co-cultures of primary AML cells or CB samples independently with UCART123 cells and second as co-cultures of a mixture of 1:1 AML and CB with UCART123 cells. The data does not show higher toxicity against normal cells when co-culture together suggesting that the cell death effect is directed to the leukemia cells expressing CD123.

Figure 1B – in vitro killing assay against MOLM13 seems to result in a max. killing plateau at ~60% dead cells at higher e:t ratios despite the fact that MOLM13 are CD123 high. Is this a technical artefact or a limitation of UCART123 killing abilities?

We believe it is a technical artifact rather than limited killing capabilities of UCART123. The data shown in **Figure 1b** has been generated after thawing of several UCART123 batches, co-cultured with MOLM13 target cells at different Effector:Target ratios. Co-cultures were maintained only overnight and the viability of remaining target cells was then evaluated by flow cytometry. This allows quantification of “whole” dead cells, but not of those having already deteriorated into debris. It is possible that higher percentages of dead cells could be obtained after longer incubation periods or when considering the amount of target cells remaining after the co-culture, but it has not been determined. However, data with primary samples shows potent anti-leukemia activity in vitro and in vivo.

Figure 2A – color code to the different AML samples is missing; Figure 2C shows high standard deviation – explanation? From the legend it is unclear if the graph shows a single CB donor with 6 replicative or several donors? In Figure 2D there are no negative values in comparison to 2C for CB – explanation? Only a single AML sample is used for comparison – how does CD123 density on this AML sample compares to other samples?

We apologize for the oversight and the lack of clarity. This has been corrected; we have added the color code to the different AML samples. The high deviation for **figure 2c** (CB cells were co-cultured with UCART123) likely has to do with the variability from donor to donor.

Figure 2c shows co-cultures of UCART123 cells with a mixture of CB/AML. For this test, 3 different CB donors were used and one AML sample (AML37). The goal of this test was to determine the selectivity of UCART123. It is likely that in this system CB cells are more vulnerable to cell death due to the culture conditions and the high percent of dead cells from the AML sample.

Figure 3C – treatment with UCART123 was started later than in 3B – why? Why ARA-C treatment was started earlier? Number of animals per group is missing.

Treatment of the PDX animals was started once chimerism was established in the mice, and that is slightly different from PDX to PDX. Ara-C was started earlier due to a technical problem when thawing the UCART123 cells on the same day Ara-C was given for PDX-AML37. However, the conclusions are similar as for PDX-AML2, Ara-C treated mice relapsed providing limited benefit on overall survival.

Figure 4 – which AML sample was used for the experiment? How does antigen density compare to other AML samples?

We apologize for the lack of details in the methodology, we used AML2. **Figure 4b** shows the difference on the expression of CD123 between normal and leukemic cells within the animals. Unfortunately, we did not include calibration beads to quantify antigen density.

Figure 4C – there seems to be an effect on the normal hematopoietic compartment in the presence of UCART123 – please provide explanation.

We have changed the presentation of Figure 4, so the differences between humanized models is better appreciated, we also show the basal levels of CD123 in each of the humanized cohorts and expanded on the manuscript's text on the interpretation for **Figure 4c** where we more clearly describe the impact on myeloid cells upon UCART123 treatment.

Reviewers' Comments:

Reviewer #2:

Remarks to the Author:

The authors addressed most of my questions.

The reviewer suggested that the authors should incorporate the key points addressed in the response to the critiques. For instance, why there is a difference in terms of toxicity of the UCARTCD123 is not as strong in the essay as previously reported. This was addressed in the rebuttal, but not included in the relevant part in the revised manuscript.

Reviewer #4:

Remarks to the Author:

The reviewer gratefully acknowledges the answer and additional data provided by the authors, which clearly improved the manuscript.

Additional comments on the presented experimental data:

Suppl. Table 1 and 2 is very helpful; however, as UCART123 effect against CD123+ cells is so heavily dependent on target density, a graphical or numerical reference to CD123 density on the different AML samples and CBC samples is regarded as essential to understand the experimental results. The figure from Cruz et al. 2018 is helpful, but should be complemented with data from the samples used in the present study.

Extended Data 1e: The extended data set is helpful to understand that there is a potential way to eliminate UCART123 if severe toxicities arise, although presented data are sparse and need to be improved. It seems that the fluorescent signal on mouse #14 and 10 is rather a signal spill over from neighboring mice. If so, reduction of UCART123 by Rituximab would only work in 50% of mice. This could be due to the chosen model, as ADCC effects are weak in humanized NSG mice. Therefore, the reviewer suggests either to strengthen this data set or to provide evidence from earlier publications that depletion by rituximab is a feasible strategy.

Statistics: With the extended data set presented now it seems that a number of data sets may not follow a normal distribution and application of parametric tests might not be appropriate – the authors should crosscheck and perhaps use non-parametric tests if normal distribution of the data sets cannot be confirmed.

Extended figure 4d does not show the level of expression, but rather frequency of CD123 expressing cells among different cell population. Legend and reference in text should be corrected accordingly. CD123 density on CD123 expressing cells in the different populations would be more informative.

The reviewer is of the opinion that the discussion needs further improvement as not all points raised by the reviewer are addressed adequately and in order to put all aspects of the experimental results presented in the correct context. Most importantly, the reviewer misses any reference to the previous clinical experience with UCART123 and its relationship to the presented experimental data. Is it the same product? Are there differences? What clinical insights in terms of safety and efficacy were obtained so far? Without any reference to earlier trials the impression arises that a completely new approach is being introduced here, which seems not to be true (or differences must be explained in more detail).

Some additional points:

From the experimental data it remains unclear, if UCART123 differentiates between low and high expressing CD123+ cells in terms of cytotoxicity and hence might spare low expressing hematopoietic precursors at all, or if this is more a kinetic effect depending on UCART123 dose and exposure time (it is understood by the reviewer that the authors believe that the latter explanation is more likely). Hence, hematopoietic toxicity would arise if UCART123 persists in treated patients – this point should be clearly addressed in the discussion and how the authors envisage handling this issue in the clinic (e.g. elimination, bridge to transplant?). In the discussion, it is now referred to the City of Hope trial (Budde et al. 2017) – however, the reviewer understands that in that trial most patients were transplanted within 1 – 2 months, hence insights into toxicities on hematopoiesis are rather limited from this study. The authors should point out in the discussion if they see UCART as a bridge to transplant or standalone therapy in that respect. If the authors consider UCART123 as a standalone therapy, their approach needs to be discussed regarding

therapeutic safety in comparison to pharmacologically controllable systems such as switchable CAR-T (Loff et al. 2020, Wermke et al. 2021) or flotetuzumab (Uy et al. 2021).. The shown density dependent cytotoxic reactivity of UCART123 might also affect treatable patient population, which might be limited to those with high CD123 expression (approx. 50 % according to the shown figure from Cruz et al. 2018). The authors should comment and if a CD123 expression threshold must be implemented as inclusion criteria in the planned clinical study. Persistence might be critical in terms of both efficacy and toxicity; however, this point is not discussed. Reference to experience from previous clinical trials with UCART123 and other UCART products would be very helpful to understand the translational value of the experimental data. The reviewer expects a rather short persistence in vivo as UCART123 are very likely highly susceptible to NK attack. Is there a multiple dosing envisioned and how does the murine scFv component of the CAR might result in ADA development which could lead to anaphylaxis (in AML the B cell compartment is most likely less compromised compared to patients with B cell derived malignancies and CD19-specific CAR-T).

It is recommended that the manuscript is carefully revised with regard to sentence structure, as errors have been introduced due to the corrections made (e.g. p. 6, middle section)

NCOMMS-20-29505-T “Allogeneic TCR $\alpha\beta$ Deficient CAR T-Cells Targeting CD123 in Acute Myeloid Leukemia (AML)”

We sincerely thank the reviewers for taking the time to carefully and critically review our manuscript. We feel we have addressed each of the concerns and believe that the manuscript is greatly improved. Please find below point by point our responses shown in red.

Reviewer #2

The authors addressed most of my questions.

The reviewer suggested that the authors should incorporate the key points addressed in the response to the critiques. For instance, why there is a difference in terms of toxicity of the UCARTCD123 is not as strong in the essay as previously reported. This was addressed in the rebuttal, but not included in the relevant part in the revised manuscript.

We thank the reviewer for the suggestion, we have added in the text more language that was in the previous rebuttal that will allow for more clarity.

Reviewer #3: report not provided

Reviewer #4:

The reviewer grateful acknowledges the answer and additional data provided by the authors, which clearly improved the manuscript.

Additional comments on the presented experimental data:

Suppl. Table 1 and 2 is very helpful; however, as UCART123 effect against CD123+ cells is so heavily dependent on target density, a graphical or numerical reference to CD123 density on the different AML samples and CBC samples is regarded as essential to understand the experimental results. The figure from Cruz et al. 2018 is helpful, but should be complemented with data from the samples used in the present study.

We have now incorporated the MFI in the supplemental information.

Extended Data 1e: The extended data set is helpful to understand that there is a potential way to eliminate UCART123 if severe toxicities arise, although presented data are sparse and need to be improved. It seems that the fluorescent signal on mouse #14 and 10 is rather a signal spill over from neighboring mice. If so, reduction of UCART123 by Rituximab would only work in 50% of mice. This could be due to the chosen model, as ADCC effects are weak in humanized NSG mice. Therefore, the reviewer suggests either to strengthen this data set or to provide evidence from earlier publications that depletion by rituximab is a feasible strategy. Statistics: With the extended data set presented now it seems that a number of data sets may not follow a normal distribution and application of parametric tests might not be appropriate – the authors should

crosscheck and perhaps use non-parametric tests if normal distribution of the data sets cannot be confirmed.

We have repeated the experiment to evaluate the ability of rituximab to eliminate UCART123 cells. We have now included the new dataset that demonstrates that treatment with rituximab led to a significant decrease of UCART123 cells, resulting in disease relapse and lack of protection when re-challenging the animals with leukemia.

Extended figure 4d does not show the level of expression, but rather frequency of CD123 expressing cells among different cell population. Legend and reference in text should be corrected accordingly. CD123 density on CD123 expressing cells in the different populations would be more informative.

We have modified extended figure 4d to show the MFI for the samples assessed, we have also included an AML sample as comparison.

The reviewer is of the opinion that the discussion needs further improvement as not all points raised by the reviewer are addressed adequately and in order to put all aspects of the experimental results presented in the correct context. Most importantly, the reviewer misses any reference to the previous clinical experience with UCART123 and its relationship to the presented experimental data. Is it the same product? Are their differences? What clinical insights in terms of safety and efficacy were obtained so far? Without any reference to earlier trials the impression arises that a completely new approach is being introduced here, which is seems not to be true (or differences must be explained in more detail).

We have modified the discussion.

Some additional points:

From the experimental data it remains unclear, if UCART123 differentiates between low and high expressing CD123+ cells in terms of cytotoxicity and hence might spare low expressing hematopoietic precursors at all, or if this is more a kinetic effect depending on UCART123 dose and exposure time (it is understood by the reviewer that the authors believe that the latter explanation is more likely). Hence, hematopoietic toxicity would arise if UCART123 persist in treated patients – this point should be clearly addressed in the discussion and how the authors envisage handling this issue in the clinic (e.g. elimination, bridge to transplant?). In the discussion, it is now referred to the City of Hope trial (Budde et al. 2017) – however, the reviewer understands that in that trial most patients were transplanted within 1 – 2 month, hence insights into toxicities on hematopoiesis are rather limited from this study. The authors should point out in the discussion if they see UCART as a bridge to transplant or standalone therapy in that respect. If the authors consider UCART123 as a standalone therapy, their approach needs to be discussed regarding therapeutic safety in comparison to pharmacologically

controllable systems such as switchable CAR-T (Loff et al. 2020, Wermke et al. 2021) or flotetuzumab (Uy et al. 2021).

The shown density dependent cytotoxic reactivity of UCART123 might also affect treatable patient population, which might be limited to those with high CD123 expression (approx. 50 % according to the shown figure from Cruz et al. 2018). The authors should comment and if a CD123 expression threshold must be implemented as inclusion criteria in the planned clinical study.

Persistence might be critical in terms of both efficacy and toxicity; however, this point is not discussed. Reference to experience from previous clinical trials with UCART123 and other UCART products would be very helpful to understand the translational value of the experimental data. The reviewer expects a rather short persistence in vivo as UCART123 are very likely highly susceptible to NK attack. Is there a multiple dosing envisioned and how does the murine scFv component of the CAR might result in ADA development which could lead to anaphylaxis (in AML the B cell compartment is most likely less compromised compared to patients with B cell derived malignancies and CD19-specific CAR-T).

It is recommended that the manuscript is carefully revised with regard to sentence structure, as errors have been introduced due to the corrections made (e.g. p. 6, middle section)

We have modified the discussion.

(Editorial note: Reviewer #4 has confidentially commented on your response to Reviewer #3 previous criticisms. In their opinion, the concerns have been adequately addressed, however, a better discussion of clinical-stage CD123-specific immunotherapies should be provided]

We have added in the discussion more information on anti-CD123 cellular therapies.

Reviewers' Comments:

Reviewer #3:

Remarks to the Author:

The additional data and statements have significantly improved the manuscript. 2 minor suggestions.

Lines 67-68: The description of CAR is somewhat confusing. May be best to elucidate each of the CAR regions, e.g. CARs consist of an antigen recognition domain, a hinge/transmembrane domain and an activation domain, with certain constructs including a costimulatory domain.

Consider revising "suicide switch" to safety switch.

NCOMMS-20-29505B “Allogeneic TCR $\alpha\beta$ Deficient CAR T-Cells Targeting CD123 in Acute Myeloid Leukemia”

We sincerely thank the reviewers for taking the time to review our manuscript carefully and critically. Please find below point by point our responses shown in red.

Reviewer #3 (Remarks to the Author):

Lines 67-68: The description of CAR is somewhat confusing. May be best to elucidate each of the CAR regions, e.g., CARs consist of an antigen recognition domain, a hinge/transmembrane domain and an activation domain, with certain constructs including a costimulatory domain.

Consider revising “suicide switch” to safety switch.

We thank the reviewer for the suggestions, we have modified the description of the CAR, and we have changed the term “suicide switch” to “safety switch”.